# Benefits of Physical Activity during and after Thyroid Cancer Treatment on Fatigue and Quality of Life: A Systematic Review

**DOI:** 10.3390/cancers14153657

**Published:** 2022-07-27

**Authors:** Margherita Ferrante, Giulia Distefano, Carlo Distefano, Chiara Copat, Alfina Grasso, Gea Oliveri Conti, Antonio Cristaldi, Maria Fiore

**Affiliations:** Department of Medical, Surgical and Advanced Technologies “G.F. Ingrassia”, University of Catania, 95123 Catania, Italy; marfer@unict.it (M.F.); distefano-giulia@virgilio.it (G.D.); carlodistef@gmail.com (C.D.); ccopat@unict.it (C.C.); agrasso@unict.it (A.G.); olivericonti@unict.it (G.O.C.); antonio.cristaldi81@gmail.com (A.C.)

**Keywords:** physical activity, physical exercise, thyroid cancer, quality of life, wellness

## Abstract

**Simple Summary:**

The usefulness of physical activity in a preventive key is no longer in question, whereas sports therapy is assuming an increasingly important role in cancer rehabilitation. This review provides an overview on the effects of physical activity on fatigue, quality of life (QoL) and health-related quality of life (HRQoL) in patients with thyroid cancer diagnosis both during and after treatment, with a focus on sex. We found that the level of confidence in the available evidence is very low. Future studies are needed to understand which training programs are optimal, both in terms of beneficial effects and to avoid potential adverse responses, in addition to focusing on gender differences.

**Abstract:**

Background: Several epidemiological studies have provided evidence of the usefulness of physical activity for cancer prevention, increased survival and quality of life (QoL), but no comprehensive review is available on the effects on thyroid cancer. The present systematic review provides an overview of the effects of physical activity on fatigue, QoL and health-related quality of life (HRQoL) in patients with thyroid cancer diagnosis both during and after treatment, with a focus on sex. Methods: A literature search was conducted in the PubMed and Scopus databases. We included studies investigating the impact of physical activity during and after thyroid cancer treatment, including fatigue, QoL and/or HRQoL among the outcomes. Review articles, conference papers, short communications and articles written in a language other than English articles were excluded. Study selection followed the preferred reporting items for systematic reviews and meta-analyses guidelines (PRISMA). Two reviewers independently selected the studies and assessed their eligibility. The same two reviewers independently screened studies, extracted data and assessed the risk of bias. Outcomes of interest were fatigue, QoL and HRQoL. The Newcastle–Ottawa Scale was used to assess the quality of the selected studies. We compared the outcomes between groups of patients with subclinical hyperthyroidism undergoing a physical activity program and a sedentary group, evaluating the possible presence of sex differences. Results and Discussion: We found five studies eligible for inclusion in our review; only two were prospective studies including an exercise training program. One study was a quasi-experimental study with a non-equivalent control group. Three out of five studies comprised low-quality evidence with a high risk of bias. Conclusion: The level of confidence in the available evidence is very low. A close association between physical activity and fatigue, QoL and HRQoL in patients with thyroid cancer diagnosis with a focus on sex can neither be supported nor refuted. Future studies are needed to understand which training programs are optimal, both in terms of beneficial effects and to avoid potential adverse responses, in addition to focusing on gender differences. The protocol of this systematic review was registered with PROSPERO: CRD42022322519.

## 1. Introduction

Thyroid cancer is one of the most frequent endocrine cancers worldwide [1,2]. Its incidence has considerably increased in Italy, as in many other countries, especially among women [3,4]. Globally, rates of thyroid cancer are 10.2 per 100,000 in women and 3.1 per 100,000 in men. In Italy, the incidence is very high in both men and women, ranging from 7.4 to 18.9 diagnoses per 100,000 inhabitants per year [5]. The increase in the thyroid cancer incidence observed in recent decades in most Western countries is largely due to the improvement in diagnostic methods and possibly to changes in the prevalence of risk factors [6,7]. In particular, papillary thyroid cancer (PTC) incidence has increased at an alarming rate, a phenomenon that many attribute to the advancement in diagnostic tools and increased access to screening [8]. Although this increase is occurring primarily in cancers measuring less than one cm in size, there is also a wide range of PTCs of all sizes and stages [9].

Differences between the sexes with respect to the incidence of thyroid cancer, its natural history and its prognosis are well established [10]. Dietary and environmental factors do not appear to play a role in sex differences in thyroid neoplasms, whereas genetic and hormonal factors could explain this difference. In particular, such differences could be explained by hormonal exposure, especially with regard to the role of estrogens and estrogen receptors in thyroid tumorigenesis, reprogramming and progression [10].

In addition, after a total or partial thyroidectomy, patients are treated with substitutive hormone levothyroxine, which suppresses serum thyrotropin levels and causes subclinical hyperthyroidism with a prevalence of up to 16%. Subclinical hyperthyroidism is influenced by sex, concomitant disease, age, etc. Eighty percent of patients with marked subclinical hyperthyroidism not only exhibit changes in body mass and bone mineral density but also complain of fatigue, weakness and tiredness, which compromise daily activities, contributing to a decrease in quality of life [11]. Although thyroid cancer is more common in women, there are few studies in the literature that stratify the issue by sex and/or gender. Moreover, the terms “sex” and “gender” are often mistakenly considered synonymous. However, sex and gender are not the same thing. Sex is defined by the human genotype and refers to biological differences between men and women. Gender is a fluid concept shaped by self-perception, social constructs, and culturally determined attitudes and expectations of men and women. Therefore, researchers should take into account the Sex and Gender Equity in Research (SAGER) guidelines of the European Association of Science Editors (EASE) in reporting data (EASE website: http//www.ease.org.uk/pubblications/sex-and-gender, accessed on 4 July 2022).

Physical activity has been hypothesized to positively influence the risk of various cancers, as well as quality of life during and after thyroid cancer treatment through different mechanisms. These include an effect on endogenous sex steroids and metabolic hormones, insulin sensitivity and chronic inflammation. Several emerging pathways related to oxidative stress, DNA methylation, telomere length, immune function and the gut microbiome may be considered mechanisms involved in prevention of cancer risk [12]. In addition, the relationship between physical activity and health-related quality of life, in general, differs between men and women. Yi-Hsueh Liao et al. found that although older women were more physically active than older men, older men with higher levels of physical activity had better HRQoL than older women [13].

Many observational and clinical studies have demonstrated that physical activity is associated with health benefits after breast and prostate cancer diagnosis [14,15,16,17,18,19,20], whereas few studies have examined the effects on thyroid cancer [21,22,23,24,25].

In particular, the association between thyroid carcinoma follow-up and quality of life has been analyzed in recent years with the aim of improving supportive therapy to increase survival rates with optimal quality of life. It is well known that physical activity is associated with improved physical and mental health status, as well as well-being in both diseased and non-diseased people [26]. Moreover, the association between physical activity and cancer patients is clear. In particular, physical activity achieves good results in terms of improving risk factors associated with cancer prognosis and progression, and it is also well tolerated during and after treatment without adverse events [27]. However, associations with aspects of physical activity, such as average hours exercised per week or weekly energy expenditure have yielded inconsistent results [28].

To the best of our knowledge, a systematic review of recently published data about the effects of physical activity on fatigue and/or quality of life in thyroid cancer patients has not been performed to date. Thus, in this study, we performed a systematic review to provide an overview of the effects of physical activity on fatigue, quality of life (QoL) and health-related quality of life (HRQoL) in patients with DTC diagnosis both during and after treatment, with a focus on sex.

## 2. Materials and Methods

We conducted this systematic review according to a protocol registered with PROSPERO: CRD42022322519. The protocol was not published in any peer-reviewed journal. This review was written and conducted according to the preferred reporting items for systematic review and meta-analyses (PRISMA) statement [29].

### 2.1. Inclusion Criteria

We considered cross-sectional studies/surveys and prospective studies of adult (age ≥ 18 years) patients with thyroid cancer diagnosis and whose previous level of physical activity was known. We considered all studies investigating the impact of physical activity during and after thyroid cancer treatment. To be included, a study had to use a defined outcome related to fatigue, quality of life and/or health-related quality of life.

### 2.2. Exclusion Criteria

Reviews, conference papers, short communications, editorials and articles written in a language other than English were removed from search results, either manually or using filters in the database.

### 2.3. Information Sources and Search Strategy

This review was based on an electronic literature search of the PubMed and Scopus databases between 2001 and 2021.

The following keywords were selected and combined according to search strategy: [“thyroid cancer” OR “thyroid carcinoma” OR “thyroid neoplasm”] AND [“physical activity” OR “workout” OR “physical exercise”] AND [“quality of life” OR “wellness” OR “fatigue”].

We additionally included any study related to the identified topic by reviewing references cited in screened papers that met our selection criteria but were missed by the keyword search criteria.

### 2.4. Selection Process, Data Collection Process and Effect Measures

Two authors (D.C. and D.G.) carried out the initial selection of articles independently according to the title and abstract. When an abstract was not available, the full text was obtained and evaluated. The same two authors screened full-text articles for inclusion and independently extracted the following data: author, year, place, study design, exercise type, sample size, patient characteristics, questionnaire/scale, outcome (scores) at baseline and at follow-up, age and sex when available (fatigue, HRQoL and QoL). Any identified inconsistencies were discussed by reviewers, and consensus was reached through discussion with a third author (F.M.). Articles that did not meet the selection criteria were excluded from the review, and reasons for exclusion were registered.

### 2.5. Synthesis Methods

The studies considered in the review were heterogeneous in terms of interventions, setting, study design and outcome measures, so meta-analyses could not be performed. We tabulated the main characteristics and results of each study to facilitate identification of patterns in the data and to provide a clear summary of the outcomes.

### 2.6. Evaluation of Study Quality

The authors D.C., D.G. and F.M. also performed quality assessment of all included studies; information was compared, and a consensus was established. To assess the quality of the included studies, we used the Quality Assess Scale-Newcastle Ottawa Statement (NOS) Manual [30]. The NOS scale provides a checklist of items for judging the risk of bias in the included studies including the following items: selection, comparability and outcome. In particular, it allocates a maximum of nine and ten stars for cohort and cross-sectional studies, respectively. In this review, studies with seven or more stars were considered high-quality studies, and those with six of fewer stars were classified as low-quality studies.

## 3. Results

The first phase of the search identified 297 potentially eligible studies. After screening titles and abstracts, 36 studies were considered relevant for full-text review. Of these, we excluded 33 studies that did not meet our selection criteria, and one study was a duplicate. In addition, we included two studies identified by reviewing the references cited in the screened articles that met our selection criteria but missed the keywords search criteria [23,25]. A total of five publications were included in this systematic review [21,22,23,24,25] (Figure 1).

The characteristics of the studies and of the patients examined, including outcomes, are shown in Table 1.

Included studies were published in medical journals in the field of endocrinology, which covers research on thyroid disease. The sample size varied between 43 patients from Korea [25], and 2760 from China [24], with a similar proportion of female and male participants. With respect to methodological aspects, two studies were prospective studies [21,22], two were cross-sectional studies [23,24] and one was a quasi-experimental study with a non-equivalent control group [25].

Furthermore, two studies recruited sedentary subjects and considered only DTC [21,22], and one study considered only papillary thyroid tumors and recruited subjects lacking regular exercise [25]. Two studies [23,24] recruited all histological types, but only one stratified the results by histological type [24]; both studies investigated the frequency of physical activity reported through a specific questionnaire. Wang et al. reported results stratified by level of physical activity [24].

Two studies involved training composed of 60 min of aerobic activity performed on a treadmill under the supervision of a physical education instructor and stretching exercises twice a week for a total of 12 weeks [21,22]. Alhasheni et. al. evaluated physical activity using the International Physical activity Questionnaire-7 day (IPAQ-7), which measures various levels and intensities of physical activity (e.g., vigorous, moderate or walking) [23]. Wang et al. used self-reported physical activity information obtained with the question,“How many days per week do you have 30 min of moderate physical activity (such as brisk walking and doing housework)?” [24]. Kim et al. used a home-based exercise program comprising 12 weeks of aerobic exercise (walking for 3–5 days a week for at least 150 min a week), resistance exercise (upper- and lower-body exercise twice a week, with more than two sets per session) and flexibility exercise (5 min for 12 weeks before and after aerobic and resistance exercises) [25].

Critical appraisal verification was carried out concerning 8 and 7 items for the prospective and cross-sectional study designs, respectively (Newcastle–Ottawa scale). All items had a score value of one, except the item “comparability”, which had a double score value. Three out five studies obtained a score ≤ 6, corresponding to a low-quality evidence publication and a high risk of bias. None of the prospective studies [21,22,25] had a score for the “comparability” and “assessment of the outcome” items. The authors of the cross-sectional studies [23,24] did not describe “response rate” and “assessment of the outcome” items; therefore, we assigned scores of 0 (Table 2 and Table 3).

Vigario P. et al. (2011) [21] investigated the effects of physical activity on body composition (lower-extremity muscle mass) and fatigue perception in patients receiving thyrotropin-suppressive therapy for differentiated thyroid carcinoma (DTC). The authors carried out a prospective study in which patients with subclinical thyrotoxicosis (scTox) were randomized in a non-blinded fashion to either participate or not participate in a physical activity program. Participants and non-participants in the physical activity program were designated as scTox-Tr and scTox-Sed, respectively. After a period of 3 months of training or observation, the measures of body composition, fatigue perception and cardiopulmonary function that had been performed at baseline were repeated. Concerning fatigue, patients with scTox-Tr had reduced median values in total Chalder Fatigue Scale scores (higher scores indicate greater fatigue) (30 vs. 17), as well as muscular pain (6 vs. 3) and fatigue qualification (5 vs. 2). Conversely, the median Chalder Fatigue Scale score increased in scTox-Sed patients (27 vs. 32) in comparison to baseline values, and no differences were found concerning muscular pain and fatigue qualification (Table 1). The authors do not discuss the limitations of their study [21].

Vigario P et al. (2014) [22] investigated the effect of a supervised workout program on quality of life (QoL) in people with DTC following TSH-suppressive therapy with a replacement hormone (levothyroxine). First, the authors performed a cross-sectional study to compare the quality of life (QoL) and the health-related quality of life (HRQoL) between subclinical hyperthyroidism (SCH) (31 women, 2 men) and euthyroid subjects (42 women, 7 men). Secondly, in the prospective phase of the survey, patients with DTC diagnosis were randomized in a non-blinded approach to either participate (SCH-Tr = trained people) or not participate (SCH-Sed = untrained people) in a monitored exercise training workout. After ninety days of training, all baseline valuations were repeated. SCH-Tr patients had significantly improved median physical (median variation: 2.6), psychological (median variation: 1.3), social (median variation: 1.3) and environmental (median variation: 1.4) WHOQOL-Bref scores (higher scores indicate better quality of life, ranging from 4 to 20). Median scores on the four WHOQOL-Bref dimensions were positive, indicating a beneficial effect of exercise training on patients’ quality of life. In contrast, SCH patients showed no difference in either median scores or variations of WHOQOL-Bref dimensions. With respect to HRQoL, after three months of training, SCH-Tr patients had better median scores in all SF-36 domains (higher scores indicate better quality of life, ranging from 0 to 100) than sedentary group results for “physical function” (82.5 vs. 60.0), “role—physical” (100.0 vs. 50.0), “bodily pain” (68.0 vs. 51.0), “general health” (74.5 vs. 52.0), “vitality” (60.0 vs. 45.0), “social functioning” (75.0 vs. 62.5), “role—emotional” (100.0 vs. 33.3) and “mental health” (68.0 vs. 52.2) domains. Notably, median variations were positive for “physical function” (median variation: 10.0, *p* < 0.05), “bodily pain” (11.0, *p* < 0.07), “general health” (5.0, *p* < 0.07), “vitality” (15.0, *p* < 0.05) and “mental health” (12.0, *p* < 0.07) (Figure 2). The authors do not discuss the limitations of their study [22].

Alhashemi A et al. (2017) investigated the severity and prevalence of moderate and severe fatigue in thyroid cancer survivors. The study explored the effect of physical activity on potential predictive factors [23]. More than half of the respondents (52.5%; the authors provide sex-stratified data) reported feeling unusually fatigued within the previous week. The mean brief fatigue inventory (BFI) global fatigue score was 3.5 ± 2.4 (95% CI: 3.2–3.8) out of 10 (10 is worst), and the prevalence of moderate–severe fatigue (global BFI score: 4.1–10 out of 10) was 41.4% (84/203). All 205 (152 women) participants provided information on physical activity in the IPAQ-7 questionnaire, and the median IQR (interquartile range) of weekly minutes spent on different types of activity was as follows: vigorous, 0 (0–180); moderate, 60 (0–240); and walking, 210 (90–840). Furthermore, in an exploratory multivariable regression analysis including age (per decade increase), female sex (compared to male), being unemployed or unable to work due to disability (compared to employed individuals, students, retirees, homemakers or caregivers), clinical pathologic features of disease, intermediate or high risk level for DTC (compared to low-risk disease), fT4 level (ng/ dL), duration of five or more years since first thyroid cancer surgery (compared to shorter periods) and physical activity level (per increment of 500 MET min), it was found that increased physical activity was significantly independently associated with reduction in fatigue (i.e., for each increase in physical activity of 500 MET min per week, the BFI global fatigue score was reduced by 0.08 points (95% CI: −0.12 to −0.03; *p* = 0.002)). The authors stated some limitations of the study, such as the lack of a sex and age-matched control group, the lack of data on physical and/or mental health comorbidities, potential response bias and the lack of prospectively collected repeated measures [23].

Wang T et al. (2017) [24] evaluated health-related quality of life (HRQoL) of community thyroid cancer survivors in Hangzhou, China, and explored the important correlates defining HRQoL. The Chinese versions of the Short Form 36 Health Survey (SF-36) and the European Organization for Research and Treatment of Cancer Quality of Life Questionnaire (EORTC QLQ-C30) were used. The authors carried out a population-based survey among thyroid cancer survivors registered within the Chronic Disease Surveillance System, which compiles data on all individuals newly diagnosed with chronic disease in Zhejiang Province, China. The author recruited 965 participants (772 female, 193 male). Thyroid cancer survivors reported the highest levels of fatigue and insomnia, in addition to impaired health-related quality of life compared with the age- and sex-matched reference population (Table 1). In particular, women had lower mental component (MCS) and physical (PCS) scores than men in the 35–44 year and 65 or older age groups. In addition, factors such as completing 30 min of moderate physical activity at least 5 days a week seemed to be correlated with higher physical component scores (PCS), whereas receiving a higher dose of levothyroxine intake per day and being overweight or obese seemed to be correlated with lower scores. Furthermore, both fruit consumption and physical activity seemed to be weakly associated with PCS and MCS, whereas women and patients having undergone more surgeries were correlated with lower mental component scores. Finally, factors such as employment status, marital status, per capita disposable income, physical activity per week, fruit consumption per day and type of surgery influenced health-related quality of life scores. The authors pointed out some limitations of the study, such that the study participants did not represent the general thyroid cancer population; that because of the small sample, they did not compare the histotype subgroups; that they did not consider the disease stage of cancer; and that data were self-reported, owing to the study design [24].

Kim K et al. (2018) [25] investigated the effect of a home-based exercise program on fatigue, anxiety, QoL and immune function in thyroid cancer patients (36 women, 7 men) who had received thyroid hormone replacement after thyroidectomy. Only fatigue and quality-of-life outcomes were included in this review. Subjects in the study group completed a 12-week home exercise program, whereas no intervention was applied for 12 weeks for subjects in the control group. After 12 weeks, the same intervention was applied in the latter group. The authors described the home exercise program in a step-by-step manner. It consisted of 12 weeks of aerobic exercise (walking 3–5 days a week for at least 150 min a week), resistance exercise (upper- and lower-body exercise twice a week, more than two sets per session) and flexibility exercise (5 min for 12 weeks before and after aerobic and resistance exercises). The home-based exercise program seemed to have several benefits. For example, the fatigue level of the subjects in the experimental group decreased significantly from a pre-exercise value (mean ± SD: 4.48 ± 1.46) to a post-exercise value (mean ± SD: 3.52 ± 1.74), whereas QoL functional scale (mean ± SD: 70.51 ± 12.33 vs. 82.73 ± 10.49) and symptom scale (mean ± SD: 78.03 ± 18.28 vs. 84.85 ± 13.76) items increased after training. Overall health QoL of the experimental group (mean ± SD: 55.30 ± 18.64 vs. 72.35 ± 15.30) was improved significantly after training. The authors did not discuss the limitations of their study [25].

## 4. Discussion

The aim of this systematic review was to provide an overview of the effects of physical activity on fatigue, QoL and HRQoL in patients with thyroid cancer diagnosis both during and after treatment, with a focus on sex. Only five studies met the eligibility criteria of this systematic review; these were national studies from Rio de Janeiro (Brazil), Toronto (Canada), Hangzhou (China) and Gyeongsang (Korea). The reason for excluding most of the articles was mainly a lack or absence of information/results related to the inclusion criteria.

Thyroid neoplasms are among the most curable cancers, with high long-term survival rates [31]. Due to the increasing incidence of thyroid cancer and the lower quality of life of patients, it is important to investigate the positive effects of physical activity, the characteristics of effective physical activity programs and any factors that hinder participation in such activities, with a focus on sex and gender due to the higher incidence in women.

Although caution should be exercised in interpreting our findings due to the small number of studies included in our review, the low-quality published evidence and the high risk of bias, these findings nonetheless appear to be largely in line with the literature on the positive effects on quality of life of participants in physical activity programs. In addition, a home-based exercise program appears to play an important role in the follow-up of DTC patients, leading to a reduction in fatigue, a decrease in daily anxiety, an improvement in QoL and a boost in immune function in patients receiving thyroid hormone substitution after thyroidectomy. These findings support the use of exercise in patients undergoing thyroid cancer therapy [25]. As described in the study by Vigario P. et al. [21] included in this review, patients with exogenous subclinical hyperthyroidism in differentiated thyroid cancer who adhered to exercise training exhibited a significant decrease in complaints of fatigue. In addition, other factors influencing fatigue should be considered; individuals who were unemployed or unable to work due to disability reported significantly higher levels of fatigue [23].

Only one of five studies included in this review analyzed data considering sex. The author of that study found that effects on both PCS and MCS were influenced by sex and age. These findings may have practical implications with respect to the implementation of physical activity programs aimed at treating DTC patients. Personalized medicine requires a sex and gender view of patient as a first step toward individualizing care and potentially improving outcomes. The importance of a focus on gender medicine/prevention was demonstrated in [32]. A study by Rahbari R et al. suggests the use of high-throughput genomic and proteomic approaches in the study of thyroid cancer gender disparities, as this approach may be helpful in terms of better understanding the molecular basis for gender differences in thyroid and other cancers [10].

Moreover, the association between physical activity and health-related quality of life differs between men and women [33]. For example, Yi-Hsueh Liao et al. found that older women were more physically active than older men. However, older men with higher physical activity levels had better HRQoL scores than women [13].

The relationship between physical activity and quality of life differs between men and women. Aschebrook-Kilfoy et al. found that female sex, together with young age at time of diagnosis and lower educational attainment, was highly predictive of decreased quality of life in thyroid cancer patients [34].

Physiologic sex-based differences also exist between men and women, e.g., men have longer limb levers, stronger bones, greater muscle mass and strength and greater aerobic capacity. In addition, during endurance exercise, women’s muscles fatigue less quickly and recover more rapidly [35]. Therefore, physical activity programs in future studies should include sex-specific activities and durations.

Moreover, physical activity programs should take into account the differences in recreation activities, which increase with age. In Particular, men show more variance in social and sedentary occupations, whereas women show more variance in physical activities. It is likely that factors related to physical health and therefore mobility in old age contribute to increased variability in participation in leisure activities. Decline is likely to be an expression of disability in old age or the tendency of older people to choose less purposeful activities as they grow older, despite the importance of physical activity [33].

However, there is a need to improve the evidence on which sex and gender-specific decisions can be based. The first step toward achieving this goal is to provide more data regarding mechanistic and regulatory processes, which are comparable, and which differ between men and women.

The evidence included in this review has several limitations. First, the histotype of thyroid cancers and the clinical phases of the disease were not considered. Second, the included studies did not investigate previous comorbidity. Third, because not all selected adults completed the required questionnaire, there is a possibility of selection bias. Fourth, in all included studies, the sample was mainly made up of women. In addition, because the presence of chronic diseases was based on self-reported data, there is a possibility of recall bias.

## 5. Conclusions

In conclusion, although the level of confidence in the available evidence is very low, it is clear that physical activity has an important place in the treatment of DTC patients because of its positive effects on fatigue, QoL and HRQoL. A close association between physical activity and quality of life in patients with thyroid cancer diagnosis with a focus on sex can neither be supported nor refuted. The studies included in this review differed in terms of type of physical activity program. Not all studies used a prospective study design, and those studies that used such a study design had a small sample size. Finally, only one study stratified results by sex. Therefore, future studies are needed to understand what training programs are optimal, both in terms of beneficial effects and to avoid potential adverse responses, with a focus on sex and gender differences. Future studies should consider differences in thyroid cancer histotype, stage of cancer and any obstacles to participation in physical activity programs, in addition to stratification by sex. The results of this review could have important practical and policy implications. Our understanding of the effects of physical activity on patients diagnosed with thyroid cancer could benefit from the implementation of specific, well-designed physical activity programs that consider sex and gender differences, as well as different histological types of thyroid cancer. Finally, recognizing the potential benefits of physical activity could help health professionals to provide better supportive care.

## Figures and Tables

**Figure 1 cancers-14-03657-f001:**
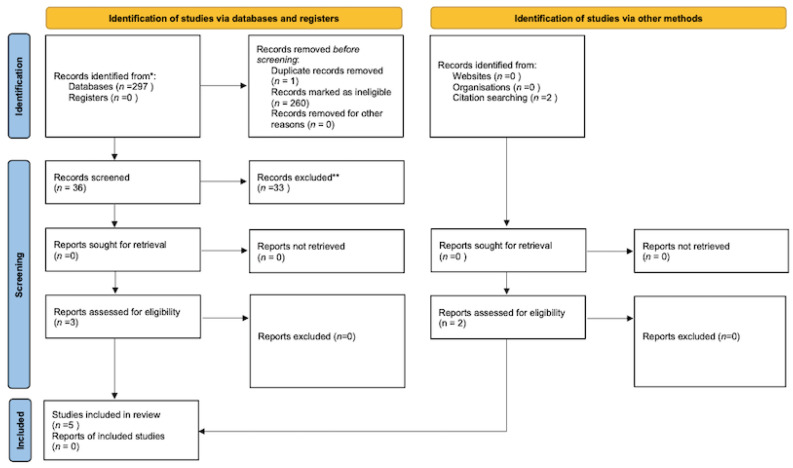
Flow chart of identification and selection studies [29]. * Consider, if feasible to do so, reporting the number of records identified from each database or register searched (rather than the total number across all databases/registers). ** If automation tools were used, indicate how many records were excluded by a human and how many were excluded by automation tools.

**Figure 2 cancers-14-03657-f002:**
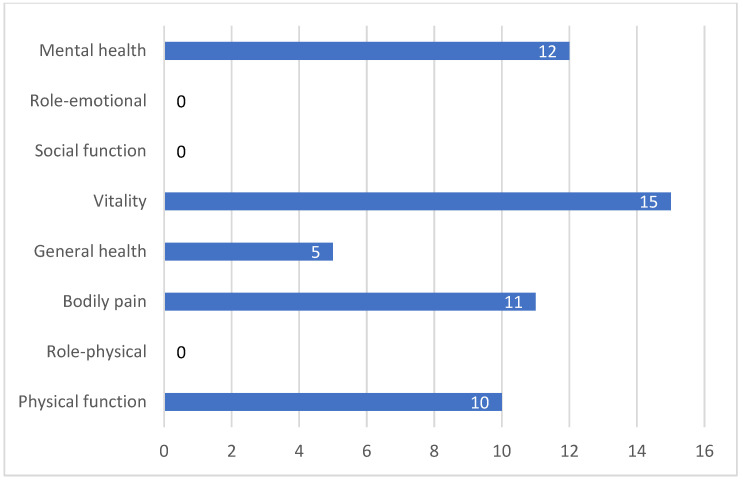
SF-36 domain median variations from baseline to 3 months of physical activity.

**Table 1 cancers-14-03657-t001:** General characteristics of the included studies.

No.	Author, Year [Reference]	Place	Study Design	Excercise Type	Sample Size	Patient Characteristics	Questionnaire/Scale ^b^	Outcome
1	P. Vigario, 2011 [21]	Rio de Janeiro, Brazil	Prospective study	Training composed of 60 min of aerobic activity performed on a treadmill under the supervision of a physical education instructor and stretching exercises twice a week for a total of 12 weeks	**Patients with scTox** ***n* = 36** (34 women, 2 men) *scTox-Tr (n = 19)* *scTox-Sed (n = 17)* ***CG n* = 48** (41 women, 7 men)	Age *scTox 48.0 (43.0–51.0) vs.* *CG 50.5 (40.2–56.0)* *(p > 0.05)* Sex (male; %) *scTox 5.6 vs. CG 14.6 (p > 0.05)* Menopause (*n*; %) *scTox 15 (41.7) vs. CG 22 (45.8)* *(p > 0.05)* BMI (kg/m^2^) *scTox 27.4(22.1–30.1) vs.* *CG 27.1 (23.4–30.3)* *(p > 0.05)* *Thyroid cancer histotype:* *DTC* Both ScTox and *CG* patients had a *sedentary lifestyle for at least 6 months before the study*	CFS	**CFS** *Baseline vs. after 3 months* scTox-Tr 30.0 (24.0–35.0) vs. 17.0 (14.0–24.0) *p* < 0.05 scTox-Sed 27.0 (23.5–34.0 vs. 32.0 (23.5–36.5) *p* < 0.05 **Muscular pain** *Baseline vs. after 3 months* scTox-Tr 6.0 (4.0–6.0) vs. 3.0 (2.0–4.0) *p* < 0.05 scTox-Sed 5.0 (4.0–6.0) vs. 5.0 (4.0–6.0) *p* > 0.05 **Qualification of fatigue** *Baseline vs. after 3 months* scTox-Tr 5.0 (4.0–6.0) vs. 2.0 (2.0–4.0) *p* < 0.05 scTox-Sed 5.0 (4.0–6.5) vs. 5.0 (3.0–6.0) *p* > 0.05
2	P. Vigario, 2014 [22]	Rio de Janeiro, Brazil	Prospective phase study	Training composed of 60 min of aerobic activity performed on a treadmill under the supervision of a physical education instructor and stretching exercises twice a week for a total of 12 weeks	**SCH *n* = 33** (31 women, 2 men) *SCH-Tr (n = 16)* *SCH-Sed (n = 17)* **EU *n* = 49** (42 women, 7 men)	Age (years): *SCH 48 vs. EU 50 (p = 0.37)* Sex (male; %) *SCH 6.1 vs. EU 14.3 (p = 0.29)* Menopause (yes; %) *SCH 42.4 vs. EU 46.9 (p = 0.25)* BMI (kg/m^2^) *SCH 26.6 vs. EU 27 (p = 0.51)* *Thyroid cancer histotype:* *DTC* SCH and EU *sedentary lifestyle for at least six months prior the beginning of the study*	**WHOQoL-Bref** Physical Psychological Social Environmental Physical Psychological Social Environmental **SF-36** Physical function Role—physical Bodily pain General health Vitality Social functioning Role—emotional Mental health Physical function Role—physical Bodily pain General health Vitality Social functioning Role—emotional Mental health	*Baseline* SCH-Tr 12.3 vs. SCH-Sed 12.6; *p* = 0.01 SCH-Tr 13.7 vs. SCH-Sed 13.3; *p* = 0.59 SCH-Tr 14.7 vs. SCH-Sed 14.3; *p* = 0.70 SCH-Tr 11.8 vs. SCH-Sed 12; *p* = 0.20 *After 3 months* SCH-Tr 15.7 vs. SCH-Sed 12.6; *p* < 0.05 SCH-Tr 14.3 vs. SCH-Sed 13.3; *p* < 0.05 SCH-Tr 14.7 vs. SCH-Sed 14.7; *p* > 0.05 SCH-Tr 15.3 vs. SCH-Sed 12.0 *p* > 0.05 *Baseline* SCH-Tr 55 vs. SCH-Sed 62.5; *p* < 0.01 SCH-Tr 50 vs. SCH-Sed 50; *p* < 0.01 SCH-Tr 46 vs. SCH-Sed 51; *p* 0.01 SCH-Tr 57 vs. SCH-Sed 52; *p* < 0.01 SCH-Tr 45 vs. SCH-Sed 40; *p* < 0.01 SCH-Tr 75 vs. SCH-Sed 50; *p* < 0.13 SCH-Tr 33.3 vs. SCH-Sed 33.3; *p* < 0.01 SCH-Tr 56 vs. SCH-Sed 44; *p* < 0.01 *After 3 months* SCH-Tr 82.5 vs. SCH-Sed 60; *p* < 0.05 SCH-Tr 100 vs. SCH-Sed 50; *p* > 0.05 SCH-Tr 68 vs. SCH-Sed 51; *p* > 0.05 SCH-Tr 74.5 vs. SCH-Sed 52; *p* > 0.05 SCH-Tr 60 vs. SCH-Sed 45.0, *p* < 0.05 SCH-Tr 75 vs. SCH-Sed 62.5 *p* > 0.05 SCH-Tr 100 vs. SCH-Sed 33.3 *p* > 0.05 SCH-Tr 68 vs. SCH-Sed 52.2 *p* < 0.05
3	A. Alhashemi, 2017 [23]	Toronto, Canada	Cross-sectional study	Various levels and intensities of physical activity (e.g., vigorous, moderate, walking)	Tot. *n* = 205 (152 women, 53 men)	Age, years (mean ± SD) *52.5 (13.5)* **Gender:** *Female:* *152/205 (74.1%)* *Male: 53/205 (25.9%)* Histotype: 91.5% DTC	BIF IPAQ-7	Mean Global Fatigue Score ± SD 3.5 ± 2.4 out of 10 (10 is worst) Prevalence of moderate–severe fatigue (score 4.1–10): 41.4% (84/203) Weekly minutes spent performing physical activity vigorous, 0 (0–180); moderate, 60 (0–240); walking, 210 (90–840)
4	T. Wang, 2017 [24]	Hangzhou, China	Population-based survey	30 min of moderate physical activity at least 5 days a week	Tot. *n* = 2755 *EG n* = 965 (772 women, 193 men) CG *n* = 1790 (Age- and sex-matched reference population)	**Gender** (*n*; %) *Male (193; 20)* *Female (772; 80)* Age (years): *49.7 ± 12.3* BMI (kg/m^2^): *23.2 ± 3.1* *EG Thyroid cancer histotype* *(n;%)* *Papillary thyroid cancer:* *(881; 92.1%)* *Follicular thyroid cancer:* *(20; 2.1%)* *Undifferentiated thyroid cancer:* *(55; 5.8%)*	SF-36 EORTC QLQ-C30	**Age (years), 14–24; female, 8** *PCS: F (49.1 ± 11.4)* *MCS:**F (46.8 ± 11.1) ^a^* **Age (years), 25–34; female, 83; male, 30** *_PCS: F (*51.7 ± 8.7*), M (*52.8 ± 8.1) *MCS:* *F (*53.8 ± 7.9), M (52.4 ± 10.1) *^a^* **Age(years) 35–44; female 166; male, 50** *PCS: F (*50.3 ± 9.8*) ^a^ M (*52.8 ± 6.6) *MCS:* *F (*51.5 ± 9.8) *^a^* M (53.7 ± 9.4) **Age (years) 45–54; female, 97; male, 65** *_PCS: F (*48.2 ± 12.0*) ^a^ M (*50.2 ± 9.3) *MCS:* *F (*49.5 ± 9.9) *^a^* M (53.6 ± 10.1) **Age (years) 55–64; female, 239; male, 32** *PCS: F (*48.6 ± 10.4*)s M (*51.1 ± 9.1) *MCS:* *F (*49.6 ± 9.0) *^a^* M (52.2 ± 8.7) **Age (years) > 65; female, 79; male, 16** *_PCS:; F (*45.4 ± 12.1*) M (*46.9 ± 10.1) *MCS:* *F (*47.8 ± 10.4) M (51.0 ± 7.9) **Global quality of life (mean ± SD)** 73.2 ± 19.2 **Functioning scale (mean ± SD)** *Role 94.5 ± 14.2* *Physical 91.9 ± 11.2* *Social 93.7 ± 14.5* *Cognitive 90.8 ± 14.6* *Emotional 91.6 ± 13.1* **Symptom scale (item) (mean ± SD)** *Fatigue 14.8 ± 18.1* *Pain 5.4 ± 12.8* *Nausea/vomiting 1.5 ± 6.7* *Dyspnea 6.9 ± 15.2* *Loss of appetite 4.9 ± 12.9* *Insomnia 12.9 ± 22.2* *Constipation 5.8 ± 16.3* *Diarrhea 2.8 ± 10.6* *Financial difficulties 5.5 ± 15.5*
5	K. Kim, 2018 [25]	Gyeongsang, Korea	Quasi-experimental study with a non-equivalent control group	A home-based exercise program: 12 weeks of aerobic exercise (walking for 3–5 days a week for at least 150 min a week), resistance exercise (upper- and lower-body exercise twice a week, more than two sets per session) and flexibility exercises (5 min for 12 weeks before and after aerobic and resistance exercises)	Tot. *n* = 43 *EG n = 22* (18 women, 4 men) *CG n = 21* (18 women, 3 men)	Age *EG 49.4 vs. CG 50.6 (p = 0.67)* Gender *EG (4 men, 18 women) vs. CG (3 men 3, 18 women) (p = 1.00)* BMI *EG 23.5 vs. CG 24.3 (p = 0.31)* *EG* *Thyroid papillary carcinoma* *EG* and *CG* *Lack of regular exercise before the study*	BIF EORTC QLQ-C30	**Fatigue** *Pre exercise vs. post exercise* *EG* 4.48 ± 1.46 vs. 3.52 ± 1.74 (*p* = 0.002) *CG* 4.37 ± 1.95 vs. 5.12 ± 1.71 (*p* = 0.001) **Functional quality of life** *Pre exercise vs. post exercise* *EG* 70.51 ± 12.33 vs. 82.73 ± 10.4 (*p* = 0.001) *CG* 71.85 ± 16.84 vs. 69.10 ± 14.79 (*p* = 0.06) **Symptom quality of life** *Pre exercise vs. post exercise* *EG* 78.03 ± 18.28 vs. 84.85 ± 13.76 (*p* = 0.01) *CG* 75.00 ± 21.08 vs. 67.46 ± 19.70 (*p* = 0.02) **Overall health quality of life** *Pre exercise vs. post exercise* *EG* 55.30 ± 18.64 vs. 72.35 ± 15.30 (*p* = 0.001) *CG* 56.35 ± 15.12 vs. 52.38 ± 18.85 (*p* = 0.23)

**Notes:** SCH = subclinical hyperthyroidism patients; SCH-Tr = trained patients; SCH-Sed = untrained patients, EU = euthyroid subjects; scTox = subclinical thyrotoxicosis; scTox-Tr = patients with subclinical thyrotoxicosis who adhered to the exercise intervention; scTox-Sed = patients with subclinical thyrotoxicosis who did not adhere to the exercise intervention; CG = control group; CFS = Chalder Fatigue Scale; WHOQOL-Bref = short version of the WHO quality of life questionnaire; SF-36 = Short Form 36 Health Survey; BFI = brief fatigue inventory; IPAQ-7 = International Physical Activity Questionnaire; EORTC QLQ-C30 = European Organization for Research and Treatment of Cancer Quality of Life Questionnaire; EG = experimental group. ^a^ = The scores of thyroid cancer patients are significantly lower than those of the reference population (*p* < 0.05). ^b^ = CFS; higher scores indicate greater fatigue; WHOQoL-Bref: higher scores indicate better quality of life, ranging from 4 to 20; SF-36: higher scores indicate better quality of life, ranging from 0 to 100; BFI: higher scores indicate the worst fatigue, ranging from 0 to 10; EORTC QLQ-C30: higher scores on the functional scale and global quality of life scores indicate better functioning and HRQoL, respectively, whereas higher scores on the symptoms scale indicate more complaints.

**Table 2 cancers-14-03657-t002:** Qualitative evaluation of prospective studies according to the Newcastle–Ottawa scale.

Author, Year	Selection				Comparability	Outcome			^a^ Total
	Representativenes of the Exposed Cohort	Selection of the Non-Exposed Cohort	Ascertainment of Outcome	Demonstration that Outcome of Interest Was Not Present at Start of Study	Comparability of Cohorts on the Basis of the Design or Analysis	Assessment of the Outcome	Was Follow-Up Long Enough for Outcome to Occur?	Adequacy of Follw-Up of Cohorts	
P. Vigario, 2011 [21]	*	*	**	*	0	0	*	*	7/9
P. Vigario, 2014 [22]	*	*	**	*	0	0	*	*	7/9
K. Kim, 2018 [25]	*	*		*	0	0	*	*	5/9

* and ** asterisks represent the stars awarded to the study for each numbered item within the NOS items. ^a^ Newcastle–Ottawa scale total score: ≤6 = low-quality studies, ≥7 = high-quality studies.

**Table 3 cancers-14-03657-t003:** Qualitative evaluation of cross-sectional studies according to Newcastle-Ottawa Scale.

Author, Year	Selection				Comparability	Outcome		^a^ Total
	Representativenes of Sample	Sample Size	Non-Respondents	Ascertainment of Exposure	The Subjects in Different Outcome Groups Are Comparable, Based on the Study Design or Analysis. Confounding Factors Are Controlled	Assessment of the Outcome	Statistical Test	
A. Alhashemi, 2017 [23]	*	*	0	**	*	0	*	6/10
T. Wang, 2017 [24]	*	*	0	**	*	0	*	6/10

* and ** asterisks represent the stars awarded to the study for each numbered item within the NOS items. ^a^ Newcastle–Ottawa scale total score: ≤6 = low-quality studies, ≥7 = high-quality studies.

## Data Availability

The data presented in this study are available on request from the corresponding author.

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
