# Peer review of "Benefits of Physical Activity during and after Thyroid Cancer Treatment on Fatigue and Quality of Life: A Systematic Review"

_cancers, 2022, doi:10.3390/cancers14153657_

Round 1
Reviewer 1 Report
The paper is really interesting considering the few studies on physical activity in thyroid cancer patients. However, before this study could be considered for publication, the following issues regarding the manuscript have to be addressed.
Title
Benefits of physical activity during and after thyroid cancer treatment: a systematic review of sex/gender differences
In my opinion, given the results of the review, inserting the focus on sex / gender differences in the title is not appropriate. In fact, although it is an argument of great interest and that will certainly have to be deepened, at the moment the few studies you have taken in consideration do not justify the focus on sex / gender differences, which could still be treated within the review, without appearing in the title.
Abstract
28-29 … “Two reviewers independently” and then “other two reviewers”
Than in Materials and Methods, line 117 - 2.4. Selection process, data collection process and effect measures, you write about two authors (DC and DG) and a third one (FM).
Please clarify.
Introduction
The introduction should be implemented with most recent studies, especially when you cite the effects of physical activity on the most studied oncological pathologies (breast and prostate cancer).
Just as an example:
De Luca V, Minganti C, Borrione P, Grazioli E, Cerulli C, Guerra E, Bonifacino A, Parisi A. Effects of concurrent aerobic and strength training on breast cancer survivors: a pilot study. Public Health. 2016 Jul;136:126-32
Soriano-Maldonado A, Carrera-Ruiz Á, Díez-Fernández DM, Esteban-Simón A, Maldonado-Quesada M, Moreno-Poza N, García-Martínez MDM, Alcaraz-García C, Vázquez-Sousa R, Moreno-Martos H, Toro-de-Federico A, Hachem-Salas N, Artés-Rodríguez E, Rodríguez-Pérez MA, Casimiro-Andújar AJ. Effects of a 12-week resistance and aerobic exercise program on muscular strength and quality of life in breast cancer survivors: Study protocol for the EFICAN randomized controlled trial. Medicine (Baltimore). 2019 Nov;98(44)
Rendeiro JA, Rodrigues CAMP, de Barros Rocha L, Rocha RSB, da Silva ML, da Costa Cunha K. Physical exercise and quality of life in patients with prostate cancer: systematic review and meta-analysis. Support Care Cancer. 2021 Sep;29(9):4911-4919.
84. …”showed a reduced risk”: Please specify what kind of risk
Materials and Methods
It would be interesting to know previous physical activity levels of the subjects recruited in the studies that you have taken in consideration for the review. Do those papers specify this aspect?
139. Please correct “less”
Conclusion
373 – only one study on stratified the results by sex/gender, but in discussion, line 329, you reported
Only two out of five studies included in this review analyzed the data considering sex.
Please clarify.
Author Response
Response to Reviewer 1 Comments
Point 1: In my opinion, given the results of the review, inserting the focus on sex / gender differences in the title is not appropriate. In fact, although it is an argument of great interest and that will certainly have to be deepened, at the moment the few studies you have taken in consideration do not justify the focus on sex / gender differences, which could still be treated within the review, without appearing in the title.
Response 1: the reviewer makes a valid point; therefore, we have rewritten the title as follows: “Benefits of physical activity during and after thyroid cancer treatment on fatigue and quality of life: a systematic review”.
Abstract
Point 2: 28-29 … “Two reviewers independently” and then “other two reviewers”
Than in Materials and Methods, line 117 - 2.4. Selection process, data collection process and effect measures, you write about two authors (DC and DG) and a third one (FM).
Please clarify.
Response 2: the reviewer makes a valid point; therefore, we have clarified that the same two authors have selected the studies, assessed their eligibility, screened studies, extracted data, assessed the risk of bias. While the third author (FM), cited in the paragraph 2.4, has been consulted in case of inconsistencies to reach consensus by discussion.
Introduction
Point 3: The introduction should be implemented with most recent studies, especially when you cite the effects of physical activity on the most studied oncological pathologies (breast and prostate cancer).
Just as an example:
De Luca V, Minganti C, Borrione P, Grazioli E, Cerulli C, Guerra E, Bonifacino A, Parisi A. Effects of concurrent aerobic and strength training on breast cancer survivors: a pilot study. Public Health. 2016 Jul;136:126-32
Soriano-Maldonado A, Carrera-Ruiz Á, Díez-Fernández DM, et al. Effects of a 12-week resistance and aerobic exercise program on muscular strength and quality of life in breast cancer survivors: Study protocol for the EFICAN randomized controlled trial. Medicine (Baltimore). 2019 Nov;98(44)
Rendeiro JA, Rodrigues CAMP, de Barros Rocha L, Rocha RSB, da Silva ML, da Costa Cunha K. Physical exercise and quality of life in patients with prostate cancer: systematic review and meta-analysis. Support Care Cancer. 2021 Sep;29(9):4911-4919.
Response 3: the reviewer makes a valid point; therefore, we have implemented the introduction with the suggested studies.
Point 4: 84. …”showed a reduced risk”: Please specify what kind of risk
Response 4: the reviewer makes a valid point; therefore, we specified that it was the risk of papillary thyroid cancer, we also clarified that some studies have investigated the protective effect of physical activity on the risk of thyroid cancer while few have dealt with its effect during or after the treatment.
Materials and Methods
Point 5: It would be interesting to know previous physical activity levels of the subjects recruited in the studies that you have taken in consideration for the review. Do those papers specify this aspect?
Response 5: the reviewer makes a valid point; therefore, we specified that we included in the review the studies carried out on subjects with thyroid cancer and whose previous level of physical activity was known.
Furthermore, in the results paragraph we specified that two studies recruited sedentary subjects (Vigario 2011, and 2014), one study recruited subjects lacking regular exercise (Kim 2018). Two studies (Wang and Alhashemi) investigate the frequency of physical activity reported through a specific questionnaire.
Point 6: 139. Please correct “less”
Response 6: Done
Conclusion
Point 7: 373 – only one study on stratified the results by sex/gender, but in discussion, line 329, you reported
Only two out of five studies included in this review analyzed the data considering sex.
Please clarify.
Response 7: we have adjusted the sentence in the discussion because due to an error we wrote two instead of one.
Reviewer 2 Report
Comments to the Author:
This is a study on benefits from physical activity on fatigue, quality of life and health-related quality of life in patients with thyroid cancer and focusing on sex differences. As physical activity might have a god effect on QoL in these patients the study is important. That being said, there are a number of limitations that need to be discussed. The introduction is not clearly written and thus difficult to understand. The language needs corrections throughout the paper. Below you can find more specific comments in the paper.
General comment:
- the text should be carefully proofread by an English-speaking person
- All abbreviations should be carefully checked, as they are not consistently used in the manuscript.
- Why is sex/gender used and not one chosen? In the Methods section only sex is used at least once. This is mentioned in the discussion but should be a part of the introduction and maybe the methods
- There are several definitions used regarding thyroid cancer patients, survivors. It is not clear which diagnosis are included and thus it is difficult to understand the results. Please consider to very consistent with diagnosis, as most studies are only on DTC.
Title:
- The title is somewhat misleading, as the paper is not about general benefits, but benefits on QoL and fatigue
- How is sex and gender defined in the study, it is confusing that both definitions are used, please specify
Simple summary:
- Lines 9-12: consider rewriting, as this information is not the main focus of the paper
Abstract:
- Line 19: the word treatment is questionable, as physical activity is not a treatment, but can be part of treatment as is might increase survival and QoL. Please correct.
- Line 20: the word information should be changed, regarding the effect on thyroid cancer?
- Line 32: the last word “treated” doesn´t seem to be correct. These patients are treatment but don´t have subclinical hypothyroidism
Introduction
- In general: there are several parts that are not clearly written, and thus the introduction is difficult to understand.
- Line 46: It would be important to make it clear if the study is about all thyroid cancers or if it is about patients with differentiated thyroid cancer. Are there any patients with ATC or MTC included in the studies?
- Lines 47-48: Is there evidence that thyroid cancer has grown more rapidly among women? Is it not a general increase? That is not clear from the studies that are referred to. An increase is evident and higher incidence among women but not a higher increase in women.
- Line 54: the word recently is questionable, as the reference is from 2014.
- Lines 58-59 lack a reference
- Lines 60-62: the sentence is difficult to understand, please rewrite
- Line 62: what does “it” refer to? Please rewrite
- Line 65: it should here be clearly stated that this part is about the treatment. It is odd to start with moreover, …
- Lines 67-68: this sentence is not possible to understand, please clarify
- Lines 68-71: to call a state of treatment associated low levels of TSH thyreotoxicosis is not correct. The reference is about DTC and these patients don´t have a concomitant thyreotoxicosis
- Line 72: cancer site risks is not correct, please clarify
- Lines 75-77: there is something lacking in the sentence, please rewrite
- Lines 78-80: it should be very clear that this is not about cancer, please rewrite
- Line 82: please clarify which cancer survivors
- Line 85: reduced risk of what? In this sentence, please use PTC as it has been explained earlier
- Lines 85-87: please check the references. Please also clarify what results have been inconsistent?
- Line 88: review on what? Please clarify.
- Line 91: is it about all thyroid cancer or only DTC? This should be clear, as there, to my knowledge, are no studies on physician activity on i.e. ATC.
Results
- General comment: as the title is about physical activity, the results section should focus on physical activity in the first place and include other information after that.
- Table 1: general comment, there is a lot of information that it is impossible to read. These are also many abbreviations that are not explained, and the Table cannot be read without reading the text or having a very good pre-understanding of the different scales/measures used. Please make the table into several tables or clarify in another way. Please explain all the questionnaires/measures and abbreviations in a footnote
- please put all the abbreviations as footnotes.
- Table 1: Prospective phase study is an unusual way of explaining a study. Is it not prospective study?
- Table 1: what is CG? Is it control group? Please explain in footnote
- Table 1: explain DTC, SF-36, EORTC and all other abbreviations in footnote
- Table 1: in many places there are p-values but not in all, please correct
- Lines 165-167: information on prospective and cross-sectional is repeated, please consider removing one of them.
- Lines 165-177: please clarify what “None of the prospective studies had a score…” means. It is difficult to understand.
- Table 2 and 3: what does the start mean? Please explain what the stars mean
- In “2.3. Evaluation of the study quality” it is written that the maximum is ten stars but in the table the total seems to be nine. Please clarify. Also include in a footnote what the numbers mean, as the table should be possible to read without reading the methods section.
- Lines 186-188: please explain what a decrease means, is that better?
- Line 193: what is the definition of thyroid neoplasm diagnosis? Is that also DTC, as in the previous study?
- Line 193: taking TSH-suppressive therapy with substitution hormone (levothyroxine): please rewrite.
- Line 198: thyroid carcinoma diagnosis, please use a clear definition
- Line 199: please rewrite, inconsistent meaning of the sentence
- Lines 201-202: what does conventional treatment mean? Was there a change to a specific treatment?
- Lines 203-205: what does the numbers mean? Is there a total score of the questionnaire that is changed?
- Lines 210-213: it would be very helpful to get a picture if these differences are significant, that information is not in the table
- Lines 213-216: what does median variations mean? Please explain.
Discussion
- General comment: in the discussion there are several studies mentioned that are not directly coupled to the studies in this review. If they will be of relevance, the information in the studies should somehow have relevance for the findings in the study.
- Lines 293-297: this part seems to fit better in the background
- Lines 298-301: please rewrite the sentence so that it is clear what the main finding in the study was.
- Lines 308-309: 2and it is also well endured during and after treatment without contrary events2: please rewrite this sentence, as it is difficult to understand.
- Lines 309-311: please clarify that this study is not about cancer. Please consider removing it, as there is so much written about cancer and all patients in the included studies are about a cancer diagnosis.
- Lines 319-321: this information should be discussed and compare to other studies, if there are some
- Lines 324-328: this fits better in the background section.
- Lines 335-336: this part does not seem to be relevant for this study, as the molecular differences are not mentioned in the included studies and these molecular differences are, at least not yet, coupled to physical activity and its effect on QoL
- Line 337-340: the findings in the study mentioned here is about elderly people without cancer and it does not seem to be relevant for the findings in this study.
- Lines 341-344: Could this study in some way be of importance for the study? Is physical activity extra important for women?
- 350-356: the same as above, how could this be of relevance for physical activity in thyroid cancer?
Conclusions
- In the conclusions it is now mentioned only that the data is inconsistent. For the reader it is evident that physical activity has an important place in the treatment of thyroid cancer. Please consider to discuss first that physical activity is important and then, as QOL in DTC women is lower than in men, maybe sex-specific issues could be the next step?
Author Response
Response to Reviewer 2 Comments
Comments to the Author:
This is a study on benefits from physical activity on fatigue, quality of life and health-related quality of life in patients with thyroid cancer and focusing on sex differences. As physical activity might have a god effect on QoL in these patients the study is important. That being said, there are a number of limitations that need to be discussed. The introduction is not clearly written and thus difficult to understand. The language needs corrections throughout the paper. Below you can find more specific comments in the paper.
General comment:
- Point 1: the text should be carefully proofread by an English-speaking person
Response 1: the reviewer makes a valid point; therefore, an English-speaking person have proofread the text
- Point 2: All abbreviations should be carefully checked, as they are not consistently used in the manuscript.
Response 2: the reviewer makes a valid point; therefore, we checked the abbreviations
- Point 3: Why is sex/gender used and not one chosen? In the Methods section only sex is used at least once. This is mentioned in the discussion but should be a part of the introduction and maybe the methods.
Response 3: the reviewer makes a valid point; therefore, we have chosen to use the word “sex” and gender when appropriate and mentioned it in the introduction.
- Point 4: There are several definitions used regarding thyroid cancer patients, survivors. It is not clear which diagnosis are included and thus it is difficult to understand the results. Please consider to very consistent with diagnosis, as most studies are only on DTC.
Response 4: the reviewer makes a valid point; therefore, we have replaced the word “survivors” with the word “patients”. In the results paragraph we specified that two studies considered only DTC (Vigario 2011, and 2014), one study considered only papillary thyroid tumors (Kim 2018). Two studies (Wang and Alhashemi) recruited all histological types but only one stratified the results by histological type (Wang).
Title:
- Point 5: The title is somewhat misleading, as the paper is not about general benefits, but benefits on QoL and fatigue
Response 5: the reviewer makes a valid point; therefore, we have rewritten the title as follows: “Benefits of physical activity during and after thyroid cancer treatment on fatigue and quality of life: a systematic review”.
- Point 6: How is sex and gender defined in the study, it is confusing that both definitions are used, please specify
Response 6:, the reviewer makes a valid point; therefore, we have chosen to use the word “sex” and gender when appropriate and mentioned it in the introduction.
Simple summary:
- Point 7: Lines 9-12: consider rewriting, as this information is not the main focus of the paper
Response 7: Done
Abstract:
- Point 8: Line 19: the word treatment is questionable, as physical activity is not a treatment, but can be part of treatment as is might increase survival and QoL. Please correct.
Response 8: Done
- Point 9: Line 20: the word information should be changed, regarding the effect on thyroid cancer?
Response 9: Done. We have replaced the word “information” with the word “effects”.
- Point 10: Line 32: the last word “treated” doesn´t seem to be correct. These patients are treatment but don´t have subclinical hypothyroidism
Response 10: Done. We have replaced the word “treated” with the word “sedentary group”.
Introduction
- Point 11: In general: there are several parts that are not clearly written, and thus the introduction is difficult to understand.
Response 11: the reviewer makes a valid point; therefore, we have reviewed the introduction.
- Point 12: Line 46: It would be important to make it clear if the study is about all thyroid cancers or if it is about patients with differentiated thyroid cancer. Are there any patients with ATC (anaplastic thyroid cancer) or MTC included in the studies?
Response 12: the reviewer makes a valid point; therefore, we have specified that two studies considered only DTC (Vigario 2011, and 2014), one study considered only papillary thyroid tumors (Kim, 2018). Two studies (Wang and Alhashemi) recruited all histological types but only one stratified the results by histological type (Wang, 2017).
- Point 13: Lines 47-48: Is there evidence that thyroid cancer has grown more rapidly among women? Is it not a general increase? That is not clear from the studies that are referred to. An increase is evident and higher incidence among women but not a higher increase in women.
Response 13: the reviewer makes a valid point; therefore, we have rewritten the phrase to clarify that there was a general increase of the thyroid cancer both in men and women but with an incidence higher among women.
- Point 14: Line 54: the word recently is questionable, as the reference is from 2014.
Response 14: the reviewer makes a valid point; therefore, we have eliminated the word “recently”
- Point 15: Lines 58-59 lack a reference
Response 15: the reviewer makes a valid point; therefore, we have included the reference number 10
- Point 16: Lines 60-62: the sentence is difficult to understand, please rewrite
Response 16: the reviewer makes a valid point; therefore, we have rewritten the sentence
- Point 17: Line 62: what does “it” refer to? Please rewrite
Response 17: the reviewer makes a valid point; therefore, we have rewritten the sentence to clarify that “it” refers to sex differences
- Point 18: Line 65: it should here be clearly stated that this part is about the treatment. It is odd to start with moreover, …
Response 18: the reviewer makes a valid point; therefore, we have replaced the word “Moreover” with the one “Additionally”
- Point 19: Lines 67-68: this sentence is not possible to understand, please clarify
Response 19: the reviewer makes a valid point; therefore, we have replaced the word “besides” with a comma
- Point 20: Lines 68-71: to call a state of treatment associated low levels of TSH thyreotoxicosis is not correct. The reference is about DTC and these patients don´t have a concomitant thyreotoxicosis
Response 20: the reviewer makes a valid point; therefore, we have replaced the word “thyreotoxicosis” with “subclinical hyperthyroidism” and we have included the reference number 11 (Batrinos ML, 2006).
- Point 21: Line 72: cancer site risks is not correct, please clarify
Response 21: the reviewer makes a valid point; therefore, we have rewritten the sentence “Physical activity has been hypothesized to influence positively the risk of different type of cancer, as well as the quality of life during and after thyroid cancer treatment, through different mechanisms.
- Point 22: Lines 75-77: there is something lacking in the sentence, please rewrite
Response 22: the reviewer makes a valid point; therefore, we have rewritten the sentences “Physical activity has been hypothesized to influence positively the risk of different type of cancer, as well as the quality of life during and after thyroid cancer treatment, through different mechanisms. These mechanisms include an effect on endogenous sex steroids and metabolic hormones, insulin sensitivity, and chronic inflammation. Several emerging pathways related to oxidative stress, DNA methylation, telomere length, immune function, and gut microbiome may be considered other mechanisms involved in cancer risk prevention”.
- Point 23: Lines 78-80: it should be very clear that this is not about cancer, please rewrite
Response 23 the reviewer makes a valid point; therefore, we have added the word “in general”
- Point 24: Line 82: please clarify which cancer survivors
Response 24: the reviewer makes a valid point; therefore, we have eliminated the words “cancer survivors”
- Point 25: Line 85: reduced risk of what? In this sentence, please use PTC as it has been explained earlier
Response 25: the reviewer makes a valid point; therefore, we have specified “thyroid cancer risk” AND used PTC
- Point 26: Lines 85-87: please check the references. Please also clarify what results have been inconsistent?
Response 26: the reviewer makes a valid point; therefore, we have removed the phrase “However, associations with aspects of physical activity, such as average hours exercised per week or weekly energy expenditure have yielded inconsistent results”
- Point 27: Line 88: review on what? Please clarify.
Response 27: the reviewer makes a valid point; therefore, we have specified that it is a systematic review about the effects of physical activity on fatigue and or quality of life in thyroid cancer patients
- Point 28: Line 91: is it about all thyroid cancer or only DTC? This should be clear, as there, to my knowledge, are no studies on physician activity on i.e. ATC.
Response 28: the reviewer makes a valid point; therefore, we have specified that they are DTC
Results
- Point 29: General comment: as the title is about physical activity, the results section should focus on physical activity in the first place and include other information after that.
Response 29: the reviewer makes a valid point; therefore, we included information about physical activity evaluation.
- Point 30: Table 1: general comment, there is a lot of information that it is impossible to read. These are also many abbreviations that are not explained, and the Table cannot be read without reading the text or having a very good pre-understanding of the different scales/measures used. Please make the table into several tables or clarify in another way. Please explain all the questionnaires/measures and abbreviations in a footnote
Response 30: the reviewer makes a valid point; therefore, we reported the abbreviations and each score meaning in a footnote.
- Point 31: please put all the abbreviations as footnotes.
Response 31: Done
- Point 32: Table 1: Prospective phase study is an unusual way of explaining a study. Is it not prospective study?
Response 32: we have eliminated the word “phase”
- Point 1: Table 1: what is CG? Is it control group? Please explain in footnote
Response 32: Done
- Point 1: Table 1: explain DTC, SF-36, EORTC and all other abbreviations in footnote
Response 32: we have explained all abbreviations in the footnote
- Point 33: Table 1: in many places there are p-values but not in all, please correct
Response 33: Done
- Point 34: Lines 165-167: information on prospective and cross-sectional is repeated, please consider removing one of them.
Response 34: we have rewritten the phrase
- Point 35: Lines 165-177: please clarify what “None of the prospective studies had a score…” means. It is difficult to understand.
Response 35: the reviewer makes a valid point; therefore, we have rewritten the phrase. Finally, the number 0 was reported in the table too.
- Point 36: Table 2 and 3: what does the stars mean? Please explain what the stars mean
Response 36: the reviewer makes a valid point; therefore, we have explained what the stars mean
- Point 37: In “2.6. Evaluation of the study quality” it is written that the maximum is ten stars but in the table the total seems to be nine. Please clarify. Also include in a footnote what the numbers mean, as the table should be possible to read without reading the methods section.
Response 37: the reviewer makes a valid point; therefore, we reviewed the sentence and included in the footnote what the numbers mean.
- Point 38: Lines 186-188: please explain what a decrease means, is that better?
Response 38: the reviewer makes a valid point; therefore, we reviewed the sentence and explained what a decrease means.
- Point 39: Line 193: what is the definition of thyroid neoplasm diagnosis? Is that also DTC, as in the previous study?
Response 39: the reviewer makes a valid point; therefore, we replaced thyroid neoplasm diagnosis with the acronym DTC.
- Point 40: Line 193: taking TSH-suppressive therapy with substitution hormone (levothyroxine): please rewrite.
Response 40: the reviewer makes a valid point; therefore, we replaced TSH-suppressive therapy with substitution hormone (levothyroxine).
- Point 41: Line 198: thyroid carcinoma diagnosis, please use a clear definition
Response 41: the reviewer makes a valid point; therefore, we replaced thyroid carcinoma diagnosis with DTC.
- Point 42: Line 199: please rewrite, inconsistent meaning of the sentence
Response 42: the reviewer makes a valid point; therefore, we have rewritten the sentence
- Point 43: Lines 201-202: what does conventional treatment mean? Was there a change to a specific treatment?
Response 43: the reviewer makes a valid point; we had already explained what the treatment consisted of then we eliminated the repetition
- Point 44: Lines 203-205: what does the numbers mean? Is there a total score of the questionnaire that is changed?
Response 44: the reviewer makes a valid point; therefore, we explained that higher scores mean greater fatigue.
- Point 45: Lines 210-213: it would be very helpful to get a picture if these differences are significant, that information is not in the table
Response 45: the reviewer makes a valid point; therefore, we have included the Figure 2
- Point 46: Lines 213-216: what does median variations mean? Please explain.
Response 46: the reviewer makes a valid point; therefore, we explained that higher scores mean better quality of life.
Discussion
- Point 47: General comment: in the discussion there are several studies mentioned that are not directly coupled to the studies in this review. If they will be of relevance, the information in the studies should somehow have relevance for the findings in the study.
Response 47: the reviewer makes a valid point; therefore, we have moved the studies not directly coupled to the studies in this review in the background
- Point 48: Lines 293-297: this part seems to fit better in the background
Response 48: the reviewer makes a valid point; therefore, we have moved this part in the background
- Point 49: Lines 298-301: please rewrite the sentence so that it is clear what the main finding in the study was.
Response 49: the reviewer makes a valid point; therefore, we have included the main finding
- Point 50: Lines 308-309: “and it is also well endured during and after treatment without contrary events”: please rewrite this sentence, as it is difficult to understand.
Response 50: the reviewer makes a valid point; therefore, we have rewritten the sentence
- Point 51: Lines 309-311: please clarify that this study is not about cancer. Please consider removing it, as there is so much written about cancer and all patients in the included studies are about a cancer diagnosis.
Response 51: the reviewer makes a valid point; therefore, we have removed the sentence
- Point 52: Lines 319-321: this information should be discussed and compare to other studies, if there are some
Response 52: the reviewer makes a valid point; unfortunately we did not find others studies
- Point 53: Lines 324-328: this fits better in the background section.
Response 53: the reviewer makes a valid point; therefore, we have moved it in the background section
- Point 54: Lines 335-336: this part does not seem to be relevant for this study, as the molecular differences are not mentioned in the included studies and these molecular differences are, at least not yet, coupled to physical activity and its effect on QoL
Response 54: the reviewer makes a valid point; therefore, We consider this study an important suggestion for future studies
- Point 55: Line 337-340: the findings in the study mentioned here is about elderly people without cancer and it does not seem to be relevant for the findings in this study.
Response 55: the reviewer makes a valid point; We considered this study important because it provides information that may be useful for future studies
- Point 56: Lines 341-344: Could this study in some way be of importance for the study? Is physical activity extra important for women?
Response 56: the study provides important information because due to the fact that women are less likely or have more difficulty exercising, a different approach may be required to convince them to benefit from the positive effects of physical activity even if they are diagnosed with cancer.
- Point 57: 350-356: the same as above, how could this be of relevance for physical activity in thyroid cancer?
Response 57: the reviewer makes a valid point; We cited this research because this information may be useful in the implementation of future physical activity programs aimed at treating these individuals
Conclusions
- Point 58: In the conclusions it is now mentioned only that the data is inconsistent. For the reader it is evident that physical activity has an important place in the treatment of thyroid cancer. Please consider to discuss first that physical activity is important and then, as QOL in DTC women is lower than in men, maybe sex-specific issues could be the next step?
Response 58: the reviewer makes a valid point; therefore, we have discussed first that physical activity is important
Reviewer 3 Report
In this systematic review, the authors evaluated the effects of physical activity on some parameters related to fatigue and quality of life during and after treatment in patients diagnosed with thyroid cancer. The topic is interesting, the manuscript is generally well written, results are clearly described and discussed. I recommend the authors an accurate language editing.
Minor specific comments
Lines 68-71. Please check the grammar, probably a verb is missing.
Line 82. Please add a comma after [11, 12, 14-17].
Line 85. Please change to “among women engaged IN regular recreational exercise”.
Line 87. Please delete “e” and replace with a comma.
Line 135. Please correct the typo in “they used”.
Line 139. Please correct the typo in “less”.
Line 144. 37 or 39 studies? There is inconsistency between the text and Figure 1. However, something is wrong as 5 studies were included by search strategy and 2 were included by reviewing references, for a total of 7 studies (Figure 1).
Table 1. Please add the full name of BIF, CG, EORTC QLQ-C30, IPAQ-7 and SF-36 in the list of abbreviations, and not within the table.
Lines 161-162. Please delete “a” before “prospective” and “cross sectional”.
Lines 165-166. There is a repetition of what previously stated.
Line 168. Please correct the typo in “except”.
Line 201. Please provide the abbreviation DTC on line 180, when you cited differentiated thyroid carcinoma/cancer.
Line 250. The full names of MCS and PCS should be provided here and not on lines 253 and 256.
Line 287. These acronyms have been already provided.
Line 313. Please see the previous comment.
Line 318. Please use the acronym DTC.
Author Response
Response to Reviewer 3 Comments
In this systematic review, the authors evaluated the effects of physical activity on some parameters related to fatigue and quality of life during and after treatment in patients diagnosed with thyroid cancer. The topic is interesting, the manuscript is generally well written, results are clearly described and discussed. I recommend the authors an accurate language editing.
Minor specific comments
- Point 1: Lines 68-71. Please check the grammar, probably a verb is missing.
Response 1: Done
- Point 2: Line 82. Please add a comma after [11, 12, 14-17].
Response 2: Done
- Point 3: Line 85. Please change to “among women engaged IN regular recreational exercise”.
Response 3: Done
- Point 4: Line 87. Please delete “e” and replace with a comma.
Response 4: the reviewer makes a valid point; therefore, we have modified the whole sentence according to the requests of the other two reviewers
- Point 5: Line 135. Please correct the typo in “they used”.
Response 5: Done
- Point 6: Line 149. Please correct the typo in “less”.
Response 6: Done
- Point 7: Line 144. 37 or 39 studies? There is inconsistency between the text and Figure 1. However, something is wrong as 5 studies were included by search strategy and 2 were included by reviewing references, for a total of 7 studies (Figure 1).
Response 7: the reviewer makes a valid point; therefore, we have clarified the inconsistency and updated Figure 1
- Point 8: Table 1. Please add the full name of BIF, CG, EORTC QLQ-C30, IPAQ-7 and SF-36 in the list of abbreviations, and not within the table.
Response 8: the reviewer makes a valid point; therefore, we have included a list of abbreviations after the abstract
- Point 9: Lines 161-162. Please delete “a” before “prospective” and “cross sectional”.
Response 9: Done
- Point 10: Lines 165-166. There is a repetition of what previously stated.
Response 10: the reviewer makes a valid point; therefore, we have erased the repetition
- Point 11: Line 168. Please correct the typo in “except”.
Response 11: Done
- Point 12: Line 201. Please provide the abbreviation DTC on line 180, when you cited differentiated thyroid carcinoma/cancer.
Response 12: Done
- Point 13: Line 250. The full names of MCS and PCS should be provided here and not on lines 253 and 256.
Response 13: Done
- Point 14: Line 287. These acronyms have been already provided.
Response 14: Done
- Point 15: Line 313. Please see the previous comment.
Response 15: Done
- Point 16: Line 318. Please use the acronym DTC.
Response 16: Done
Round 2
Reviewer 2 Report
Thank you for revising the manuscript. I have no furher comments.